# A Search for Anti-*Naegleria fowleri* Agents Based on Competitive Exclusion Behavior of Microorganisms in Natural Aquatic Environments

**DOI:** 10.3390/pathogens10020142

**Published:** 2021-02-01

**Authors:** Pichet Ruenchit, Narisara Whangviboonkij, Hathai Sawasdipokin, Uraporn Phumisantiphong, Wanpen Chaicumpa

**Affiliations:** 1Department of Parasitology, Faculty of Medicine Siriraj Hospital, Mahidol University, Bangkok 10700, Thailand; narisara.wha@mahidol.ac.th (N.W.); hathai.noc@mahidol.ac.th (H.S.); 2Department of Clinical Pathology, Faculty of Medicine Vajira Hospital, Navamindradhiraj University, Bangkok 10300, Thailand; uraporn@nmu.ac.th; 3Center of Research Excellence on Therapeutic Proteins and Antibody Engineering, Department of Parasitology, Faculty of Medicine Siriraj Hospital, Mahidol University, Bangkok 10700, Thailand; wanpen.cha@mahidol.ac.th

**Keywords:** amphizoic *Naegleria fowleri*, competitive exclusion, primary amoebic meningoencephalitis (PAM), *Pseudomonas aeruginosa*, pyocyanin, aquatic environment

## Abstract

*Naegleria fowleri* causes deadly primary amoebic meningoencephalitis (PAM) in humans. Humans obtain the infection by inhaling water or dust contaminated with amebae into the nostrils, wherefrom the pathogen migrates via the olfactory nerve to cause brain inflammation and necrosis. Current PAM treatment is ineffective and toxic. Seeking new effective and less toxic drugs for the environmental control of the amoeba population to reduce human exposure is logical for the management of *N. fowleri* infection. On the basis of the concept of competitive exclusion, where environmental microorganisms compete for resources by secreting factors detrimental to other organisms, we tested cell-free culture supernatants (CFSs) of three bacteria isolated from a fresh water canal, i.e., *Pseudomonas aeruginosa*, *Pseudomonas otitidis*, and *Enterobacter cloacae*, were tested against *N. fowleri*. The CFSs inhibited growth and caused morphological changes in *N. fowleri*. At low dose, *N. fowleri* trophozoites exposed to *P. aeruginosa* pyocyanin were seen to shrink and become rounded, while at high dose, the trophozoites were fragmented. While the precise molecular mechanisms of pyocyanin and products of *P. otitidis* and *E. cloacae* that also exert anti-*N. fowleri* activity await clarification. Our findings suggest that *P. aeruginosa* pyocyanin may have a role in the control of amphizoic *N. fowleri* in the environment.

## 1. Introduction

*Naegleria fowleri* is an amphizoic amoeba that inhabits fresh water and soil [1]. This protozoan (colloquially “brain-eating amoeba”) is an etiological agent of primary amoebic meningoencephalitis (PAM) or amoebic encephalitis/meningitis, which is a rare but usually fatal-neurological disorder in humans [2]. Humans obtain the infection through choking water containing *N. fowleri* trophozoites into the nostril during recreational activities such as swimming, boating, rafting, or inhaling contaminated dust [1,3]. The amoeba then migrates from the nasal cavity to the brain via the olfactory nerves and cribriform plate, thereby accessing the subarachnoid space and brain, resulting in inflammation and extensive necrosis [1,3]. Clinical manifestations of PAM include severe frontal headache, fever, positive Brudzinski and Kernig signs, photophobia, confusion, seizures, and coma [4]. *N. fowleri* was first identified in the 1960s in Southern Australia where unchlorinated warm water from a river was supplied to the town through an aboveground pipeline [5,6]. Recently, nearly 500 cases of PAM have been reported from different countries of Africa, America, Asia, Australia, Europe, and Oceania [7,8], including 12 cases reported in Thailand since 1983 [9]. Onset of clinical signs and symptoms vary from 2–3 days to as long as 15 days [10]. Progression of PAM is usually rapid and unresponsive to treatment, leading to death within a few days in more than 95% of cases [10,11,12]. Accurate diagnosis of PAM is challenging due to clinical and brain-imaging similarities with other meningitides, suggesting that this parasitic infection is likely to be underreported.

Aquatic environments, such as rivers, lakes, and hot springs, are well-known natural reservoirs of *N. fowleri* [13,14]. Usually, environmental microorganisms that share overlapping niches gain competitive advantages over other organisms and microbes. Microorganisms compete for space and resources by secreting extracellular chemical substances that inhibit other microorganisms [15], leading to competitive exclusion of rival species [16,17]. For example, *Burkholderia thailandensis* inhibits the growth of *Bacillus subtilis* by secreting an antibiotic substance [18], while its growth is inhibited by its close relative *B. pseudomallei* [19]. Environmental non-O1/non-O139 *Vibrio cholerae* inhibits the growth of the O1 pandemic strain by using contact-dependent type VI secretion system (T6SS) and secreting the hemolysin A for aquatic niche competition [15]. Moreover, *Pseudomonas aeruginosa*, a free-living bacterium, displays inhibitory activity against free-living amoeba (*Acanthamoeba* spp.) [20,21], bacteria (*Streptococcus pneumoniae*, *Escherichia coli*, and *Salmonella* Typhi) [22], and fungi (*Candida* spp. and *Aspergillus fumigatus*) [23]. These observations have important ecological and clinical implications. Thus, it is logical to search for an anti-*N. fowleri* agent produced by bacteria isolated from the natural aquatic habitat that might have potential applications to control the pathogenic amoeba population.

## 2. Results and Discussion

### 2.1. Anti-N. fowleri Effect of the Cell-Free Culture Fluids of Aquatic Bacteria

On the basis of the concept of competitive exclusion by competitive interaction [16], this study explored culture supernatants of aquatic bacteria that might contain substances that are effective against the pathogenic *N. fowleri*. Three aquatic bacterial isolates were randomly selected from the aquatic bacterial collection. As shown in Figure 1A, KP-01 cultured in Luria–Bertani (LB) medium at 37 °C overnight in LB broth produced a cell-free supernatant (CFS) of blue-green hue with a fermented grape-like odor, while KP-14 and KP-15 isolates did not cause the CFS color to change. After a 48 h-treatment of *N. fowleri* by the CFSs, growth inhibition was found in a dose-dependent manner (Figure 1B–D). The CFSs of KP-14 and KP-15 at 1.20 mg mL^−1^ and KP-01 at 2.40 mg mL^−1^ inhibited the growth of *N. fowleri* by approximately 50% (Figure 1B–D). Using Probit analysis, the CFSs of KP-01, KP-14, and KP-15 demonstrated *N. fowleri* inhibitory concentration 50 (IC_50_) at 1.6, 1.3, and 1.5 mg mL^−1^, respectively.

CFS of KP-01 isolate at 1.6 mg mL^−1^ exerted in vitro anti-*N. fowleri* activity during testing in a T 25-cm^2^ flask. Viable *N. fowleri* treated with the KP-01 CFS was approximately 5 × 10^5^ cells at 48 h-post-treatment, while viable trophozoite numbers in both LB and Nelson’s media increased to 1.27 × 10^7^ (25-fold difference; *p* < 0.001) (Figure 2A). These results conformed to the data of the proliferation assay, in which the percent survival of the amoebae after treatment with the KP-01 CFS was reduced significantly compared to the LB medium treatment (*p* < 0.001) (Figure 2B). The trophozoites exposed to the KP-01 CFS became rounded, small, and did not attach to the plastic surface of the flask (Figure 3). This morphological change was similar to the trophozoites exposed to amphotericin B reported previously [24]. The trophozoite subcellular organelles burst, which was seen as small black particles in the culture medium. Attachment of the remained viable trophozoites treated with the KP-01 CFS was not observed at 0 and 24 h after CFS removal and fresh medium replacement at 48 h-post-treatment as shown in Appendix A. This event was also found in the floating trophozoites after CFS removal. The floating trophozoites that were treated with the KP-01 CFS were not able to attach to the plastic surface of the flask when optimal condition was restored at 24 h (Appendix A). In contrast, the trophozoites exposed to LB and Nelson’s media were intact and revealed a normal amoeboid form approximately 10–30 µm in size (Figure 3).

Growth of the *N. fowleri* treated with the CFSs of KP-14 and KP-15 was also inhibited, as shown by the trypan blue exclusion method (Figure 2A) and cell proliferation assay (Figure 2B). At 48 h post-treatment, the viable numbers of *N. fowleri* exposed to the CFSs of KP-14 and KP-15 increased approximately six- and fourfold compared to the original input number, respectively. In contrast, the control groups treated with LB and Nelson’s media yielded 1.27 × 10^7^ trophozoites mL^−1^, a 25-fold increase from the beginning number (Figure 2A). The percent survival of the amoebae treated with CFSs of KP-14 and KP-15 were statistically different from that of LB treatment (*p* < 0.001) (Figure 2B). The microscopic appearance of the trophozoites treated with the CFSs of KP-14 and KP-15 were similarly changed as for the KP-01 CFS-treated counterpart (Figure 3). It is to be noted that the CFS of KP-01 was the most effective anti-*N. fowleri* among the three CFS preparations (Figure 2A). The discovery that the CFSs from the three unknown bacteria each had different appearances and demonstrated anti-*N. fowleri* activity to varying degrees persuaded us to investigate the bacterial species.

### 2.2. Identification of the Aquatic Bacteria That Produced Anti-N. fowleri Products

Gram’s staining and microscopic observation of the three aquatic isolates (KP-01, KP-14, and KP-15) showed a Gram-negative rod appearance (Figure 4A). On LB agar, the KP-01 formed blue-green flat colonies with a grape-like odor; colonies of the KP-14 were light-green and flat, whereas those of the KP-15 were opaque and convex (Figure 4B), both without typical odor. All three bacteria grew aerobically on MacConkey agar as non-lactose fermenters but could not grow on thiosulfate citrate bile salt sucrose (TCBS) agar (data not shown), verifying the characteristics of Gram-negative bacteria beyond the Vibrionaceae family. To identify the species of these aquatic bacteria, biochemical profiles were determined using the Automated MicroScan system. The KP-01 biochemical profile, shown in Table 1, was positive to citrate, acetate, arginine dihydrolase, maltose, nitrate, oxidase, and oxidative fermentation (OF)/glucose tests. Thus, this bacterial isolate was identified as *Pseudomonas aeruginosa* with 99.99% probability. The KP-14 gave positive results for citrate, arginine dihydrolase, maltose, oxidase, and oxidative fermentation (OF)/glucose tests, and was identified as *Pseudomonas putida* with 99.57% probability. The KP-15 was found to be *Enterobacter cloacae* (78.94% probability) on the basis of the positive results for glucose, raffinose, inositol, citrate, 1,2,4,5-tetrachlorobenzene (CL4), sucrose, esculin, maltose, nitrate, sorbital, ornithine decarboxylase, Voges–Proskauer, ortho-nitrophenyl-β-D-galactopyranoside (OPNG), and oxidative fermentation (OF)/glucose tests. In addition, on the basis of nucleotide sequencing of the 16S ribosomal RNA (rRNA) gene, we found that the KP-01 and KP-14 isolates clearly belonged to *Pseudomonas* spp., while the KP-15 isolate belonged to *Enterobacter* spp. (Figure 5). The KP-01 was closely related to *Pseudomonas aeruginosa* (99.14% identity), while the KP-15 was closely related to *Enterobacter cloacae* (100% identity). These results were supported with a bootstrap value of 100%. However, the KP-14 was more related to *Pseudomonas otitidis* (99.71% identity) than *Pseudomonas putida*, which was different to the result of biochemical profiling. This could have been caused by a low discriminatory power of biochemical profiling in species identification between *P. putida* and *P. otitidis* since their biochemical profiles are closely similar. It was reported that phenotypic and genotypic characteristics within the *Pseudomonas* genus are closely related, resulting in misidentification, especially *P. otitidis* [25]. Ochman et al. reported that bacterial phenotypic characteristics including Gram stain results, colony morphologies, and biochemical profiles are not static, present with uncommon phenotypes compared to the reference strain, and can change with stress or evolution [26]. Since limitation of biochemical profiling depends on the discriminatory power and an update of databases, full or partial 16S rRNA gene sequencing has emerged as a useful tool for identifying phenotypically aberrant microorganisms. It showed a more objective, accurate, and reliable method for bacterial identification [27]. Therefore, the KP-14 isolate is identified as *P. otitidis* but not *P. putida* in terms of this evidence.

Among several aquatic bacterial species that are Gram-negative, *Escherichia coli*, *Xanthomonas maltophilia*, *Flavobacterium breve*, and *P. paucimobilis* have been reported to support growth of free-living amoebae [28]. *P. aeruginosa* was shown previously to exert a lethal effect on *Acanthamoeba* spp. [20,21]. These data show for the first time that *P. aeruginosa* and *P. otitidis* as well as *E. cloacae* have anti-*N. fowleri* activity.

### 2.3. Effect of Pyocyanin on N. fowleri

Saha et al. reported that almost 95% of antimicrobial inhibitory activity of *P. aeruginosa* is attributable to pyocyanin secretion [29]. Pyocyanin, chemically *N*-methyl-1-hydroxyphenazine, is a blue-green phenazine compound produced by *P. aeruginosa* that has antibiotic activity against multi-drug-resistant bacteria including *Streptococcus pneumoniae*, *Escherichia coli*, and *Salmonella* Typhi [22]. Pyocyanin also possesses anti-fungal activity against *Candida albicans* and *Aspergillus fumigatus* [23]. The KP-01 *P. aeruginosa* produced blue-green colored broth culture and colonies, indicating that the bacteria secrete pyocyanin. Therefore, pyocyanin was the first *P. aeruginosa* candidate that we tested for anti-*N. fowleri*.

*N. fowleri* trophozoite (1 × 10^4^ cells) treated with various concentrations of pyocyanin for 24 h revealed cytotoxicity in a dose-dependent manner, compared to *N. fowleri* treated with medium alone or medium containing 1% DMSO, which was used as the pyocyanin diluent (Figure 6A,B). The IC_50_ value of the pyocyanin was 23 µg mL^−1^ by Probit analysis. Moreover, pyocyanin at 150 and 300 µg mL^−1^ were as cytotoxic to *N. fowleri* as 10 µg mL^−1^ amphotericin B (*p* > 0.05) (Figure 6A,B).

By inverted microscopy, *N. fowleri* trophozoites treated with 150 and 300 µg mL^−1^ pyocyanin or 10 µg mL^−1^ amphotericin B for 24 h showed similar morphology, i.e., the trophozoites were fragmented and only the remnants were seen as small black particles in the culture medium (Figure 7). The trophozoites exposed to pyocyanin at a smaller dose, 75 µg mL^−1^, had a rounded and shrunken appearance in contrast to the trophozoites exposed to medium containing 1% DMSO diluent that showed intact amoeboid form (Figure 7). Reduction in size and rounded morphology of *N. fowleri* trophozoites after exposure to amphotericin B for 24 h has been reported previously [30].

The results of this study indicate that pyocyanin, one of the *P. aeruginosa* virulence factors, has anti-*N. fowleri* activity and verified the concept of competitive exclusion among environmental microorganisms. Although the molecular mechanism of pyocyanin and the nature of the products from *P. otitidis* and *E. cloacae* need further investigation, the *P. aeruginosa*-derived pyocyanin has potential applications for environmental control of the pathogenic free-living *N. fowleri*. However, the cytotoxicity of pyocyanin remains a concern given its deleterious effects on cells in other body systems reported in vitro and in vivo studies, including the urological [31], the respiratory [32], the cardiovascular [33], and the central nervous systems [34].

## 3. Materials and Methods

### 3.1. N. fowleri, Bacteria, and Culture Conditions

*N. fowleri* used in this study was a reference CDC VO 3081 strain from Dr. GS Visvesvara at the Centers for Disease Control and Prevention (USCDC). The amoeba was axenically cultured in Nelson’s medium supplemented with 10% heat-inactivated fetal bovine serum (FBS) (GE Healthcare Lifesciences, Logan, UT, USA) at 37 °C in T 25-cm^2^ flask (Corning, New York, NY, USA) in the secure facility of the Department of Parasitology, Faculty of Medicine Siriraj Hospital, Mahidol University, Bangkok.

Three isolates of unidentified environmental bacteria, designated KP-01, KP-14, and KP-15, were randomly selected from a culture stock of bacteria isolated from a fresh water canal in Bangkok. These bacteria were grown in Luria–Bertani (LB) broth at 37 °C with shaking aeration until the stationary phase of growth.

### 3.2. Chemical Agents

Amphotericin B was from Bharat Serums and Vaccines (Mumbai, India). Pyocyanin was from Sigma-Aldrich (Rockville, MD, USA). Stock solutions of amphotericin B and pyocyanin were prepared by dissolving the reagents in sterile distilled water and 1% dimethyl sulfoxide (1% DMSO), respectively, to 5 mg mL^−1^, and sterilized by filtering through a 0.22 µm pore-size filter (PALL, Fribourg, Switzerland). Working solutions were prepared by diluting the stock solutions in Nelson’s medium to the desired concentrations.

### 3.3. Preparation of the Cell-Free Culture Supernatants of the Environmental Bacteria

The aquatic bacteria—KP-01, KP-14, and KP-15—were grown overnight in 5 mL LB broth at 37 °C with agitation. The cultures were then centrifuged at 5000× *g*, 4 °C, for 20 min. Next, the supernatants were filtered through a 0.22 µm pore-size filter (PALL, Fribourg, Switzerland) and concentrated using Concentrator Plus (Eppendorf AG, Hamburg, Germany). Protein contents of the concentrated cell-free supernatants (CFSs) were determined using the SMART^TM^ BCA Protein Assay kit (Intronbio, Gyeonggi-do, South Korea).

### 3.4. N. fowleri Inhibition Assay

To evaluate the anti-*N. fowleri* activity of the bacterial CFSs, we harvested the amoebae cultivated in the Nelson’s medium supplemented with 10% FBS at 37 °C to logarithmic phase. The number of living trophozoites was counted using the trypan blue exclusion method [35]. A total of 10,000 *N. fowleri* living cells in 200 µL of the Nelson’s medium supplemented with 10% FBS, 100 U mL^−1^ penicillin, and 100 µg mL^−1^ streptomycin were seeded to individual wells of a 96-well microplate, and incubated at 37 °C for 24 h. After discarding the culture medium, we added various concentrations of the CFSs (200 µL of 0.15, 0.30, 0.60, 1.20, and 2.40 mg mL^−1^ per well; triplicate wells for each CFS concentration) to the wells containing the amoebae. Amoebae wells with medium alone served as negative inhibition control. The plate was kept at 37 °C for 48 h. Number of living amoebae in each treatment was determined using the CellTiter 96 Non-Radioactive Cell Proliferation Assay (Promega, Wisconsin, USA). Results are shown as mean ± standard deviation (SD) of the representative of 3 independent experiments. The half-maximal inhibitory concentration (IC_50_) of each bacterial CFS was calculated using Probit analysis [36].

### 3.5. Enumeration of N. fowleri Viable Cells and Cell Proliferation Assay

In order to confirm the anti-*N. fowleri* activity of the bacterial CFS and to clearly observe the morphology of the treated amoebae, we prepared 5 × 10^5^
*N. fowleri* trophozoites in T25-cm^2^ flask containing 5 mL of the complete Nelson’s medium supplemented with 10% FBS, 100 U mL^−1^ penicillin, and 100 µg mL^−1^ streptomycin, and incubated at 37 °C for 24 h. Each culture was treated with 1.60 mg mL^−1^ of the CFS, then incubated at 37 °C for 48 h. LB broth and Nelson’s medium alone were used as the negative controls. Viable trophozoite numbers were determined using the trypan blue exclusion method and the cell proliferation assay, as described above. Total viable amoebae were calculated and shown as mean ± SD and percent of their survival. Morphology of the *N. fowleri* treated with the bacterial CFSs and controls was observed daily under the Olympus IX70 Inverted Tissue Culture Microscope (Olympus, Tokyo, Japan) and recorded by WiFi Microscope Digital Camera Model MC4KW-G1 (Microscope X, JiangSu, China).

### 3.6. Identification of the KP-01, KP-14, and KP-15 Bacteria

The KP-01, KP-14, and KP-15 were subjected to Gram staining and then cultured on different culture media including LB, MacConkey, and thiosulfate citrate bile salt sucrose (TCBS) agar plates. To identify the bacterial species, we performed biochemical profiling using the Automated MicroScan system (Siemens Healthcare Diagnostics, New York, NY, USA) and nucleotide sequencing of 16S ribosomal RNA (rRNA) gene using 27F (5′-AGAGTTTGATCCTGGCTCAG-3′) and 1492R (5′-GGTTACCTTGTTACGACTT-3′) primers [37]. DNA was extracted from the overnight cultures using the QIAamp DNA Mini Kit (QIAGEN GmbH, Hilden, Germany). PCR reaction containing 1x AccuStart II GelTrack PCR SuperMix (Quantabio, Beverly, MA, USA), 500 nM of each primer, and 50 ng of DNA template were subjected to thermal cycler; DNA template denaturation at 95 °C for 5 min; followed by 30 cycles of DNA denaturation at 94 °C for 1 min, primer annealing at 55 °C for 2 min, and DNA extension at 72 °C for 2 min; and final extension at 72 °C for 5 min. PCR products (1466 bp) were extracted from agarose gel and submitted for sequencing. The 16S rRNA gene sequences were deposited in GenBank (accession nos. MW498239, MW498240, and MW498241 for KP-01, KP-14, and KP-15, respectively), analyzed in terms of sequence similarity by the Basic Local Alignment Search Tool for Nucleotide (BLASTN) [38], and phylogenetically analyzed using Molecular Evolutionary Genetics Analysis (MEGA) software version 10.0 [39].

### 3.7. Anti-N. fowleri Activity of Pyocyanin

Fresh *N. fowleri* culture medium (200 µL) containing different concentrations of pyocyanin (Sigma-Aldrich, Rockville, MD, USA; 4.6875, 9.375, 18.75, 37.5, 75, 150, and 300 µg mL^−1^) was added to a 96-well plate containing 1 × 10^4^ live *N. fowleri* trophozoites. Medium alone was used as negative anti-amoeba controls; medium containing 1% DMSO was used as diluent control, and medium containing 10 µg mL^−1^ amphotericin B (200 µL) was used as positive anti-amoeba control. Percent of viable amoebae were calculated: [(viable cell number of test sample/viable cell number of negative anti-amoeba sample)] × 100. Then, the IC_50_ value of pyocyanin was calculated using the data of percent amoeba survival and Probit analysis.

### 3.8. Statistical Analysis

Statistical analysis of data was performed using SPSS Statistics version 21 (SPSS, Chicago, IL, USA). One-way ANOVA followed by the Tukey’s honestly significant difference (HSD) post hoc test for multiple comparison was used to compare the mean difference between each treatment group. The results were considered significantly different when *p* < 0.05 (∗), *p* < 0.01 (∗∗), and *p* < 0.001 (∗∗∗).

## 4. Conclusions

On the basis of the concept of competitive exclusion among microorganisms in environmental niches, our study demonstrated for the first time that aquatic *P. aeruginosa* can inhibit the amphizoic *N. fowleri* amoebae through production of pyocyanin. The pyocyanin at low dose caused reduction in size and rounded morphology of amoebae, while the high dose caused the trophozoite fragmentation. In addition, the cell-free culture supernatants (CFSs) of *P. otitidis* and *E. cloacae* also showed the antiamoebic activity against *N. fowleri*. The precise molecular mechanism of the pyocyanin and the products of *P. otitidis* and *E. cloacae* that also exerted the anti-*N. fowleri* activity merit further investigation. Nevertheless, our findings suggest the potential application of *P. aeruginosa* pyocyanin to control amphizoic *N. fowleri* amoebae in environmental sources.

## Figures and Tables

**Figure 1 pathogens-10-00142-f001:**
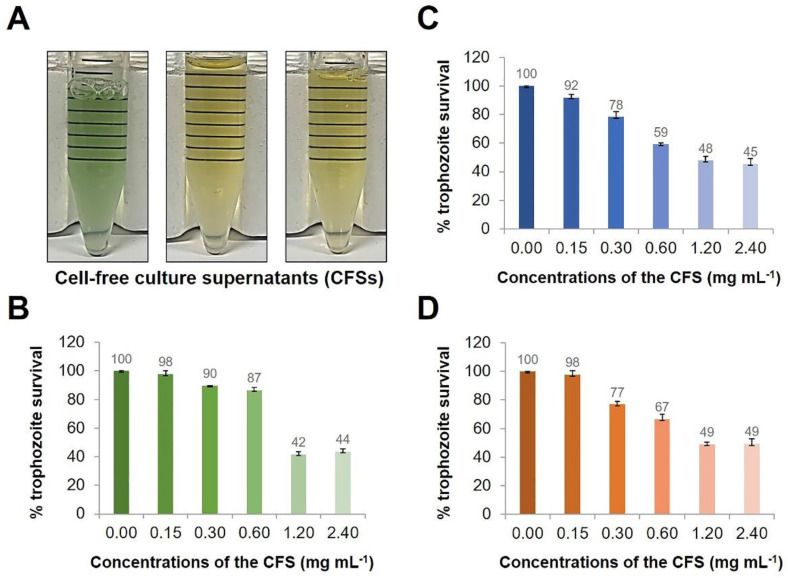
Anti-*Naegleria fowleri* activity of aquatic bacterial cell-free culture supernatants (CFSs). *N. fowleri* (1 × 10^4^ cells) in individual wells of a 96-well culture plate were treated with different concentrations (0.15, 0.30, 0.60, 1.20, and 2.40 mg mL^−1^) of CFSs of aquatic bacteria, i.e., KP-01, KP-14, and KP-15 isolates, for 48 h. (**A**) Appearance of the CFSs of KP-01 (left), KP-14 (middle), and KP-15 (right) collected at the stationary-phase of the bacterial cultures. Survival rates (%) of *N. fowleri* at 48 h post-treatment with different concentrations of the CFSs of KP-01 (**B**), KP-14 (**C**), and KP-15 (**D**). Results are shown as mean ± standard deviation (SD) of the representative of three independent experiments.

**Figure 2 pathogens-10-00142-f002:**
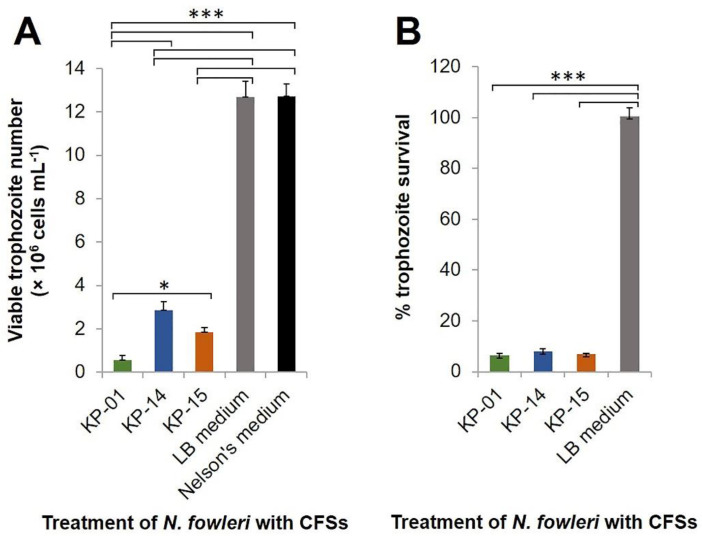
Effect of the aquatic bacterial CFSs on *Naegleria fowleri* cultivated in a T 25-cm^2^ flask. Viable numbers of the amoebae were determined 48 h post-incubation with the bacterial CFSs (1.60 mg mL^−1^), Luria–Bertani (LB) medium, and Nelson’s medium. (**A**) Numbers of viable *N. fowleri* trophozoites at 48 h post-treatment with 1.6 mg mL^−1^ of CFSs of KP-01, KP-14, and KP-15 isolates. (**B**) Percentage survival of *N. fowleri* trophozoites at 48 h post-treatment with 1.6 mg mL^−1^ of CFSs of KP-01, KP-14, and KP-15 isolates. Results are shown as mean ± standard deviation (SD) of the representative of three independent experiments. ∗, *p* < 0.05, and ∗∗∗, *p* < 0.001.

**Figure 3 pathogens-10-00142-f003:**
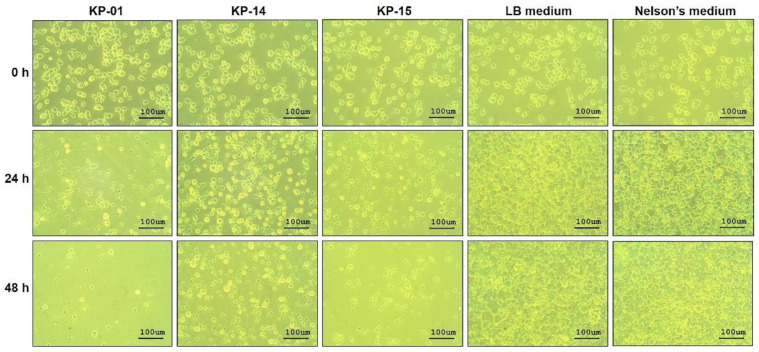
Morphology of *Naegleria fowleri* trophozoites at different time intervals after treatments with the bacterial CFSs. *N. fowleri* (5 × 10^5^ cells) in T 25-cm^2^ flask were treated with 1.60 mg mL^−1^ of CFSs of KP-01, KP-14, and KP-15, or LB or Nelson’s media alone. Morphological alterations were observed daily using an Olympus IX70 Inverted Tissue Culture Microscope (Olympus, Tokyo, Japan) at 200× magnification. Scale bars representing 100 µm size were set using Micro Capture Version 7.9.

**Figure 4 pathogens-10-00142-f004:**
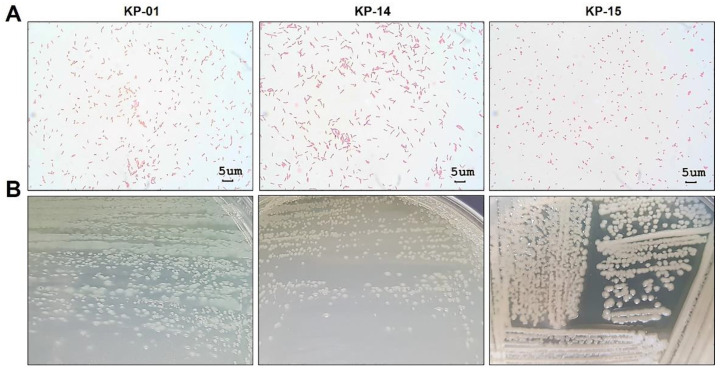
Light microscopic and colonial morphology of the three isolates of aquatic bacteria from which CFSs showed anti-*N. fowleri* activity. (**A**) The bacterial shape and arrangement observed under 1000× magnification. Scale bars representing 5 µm in size were set using Micro Capture Version 7.9. They were all Gram-negative. (**B**) Bacterial colonial morphology and pigment production on LB agar. KP-01 produced blue-green colonies; KP-14 and KP-15 revealed light-green and white colonies, respectively.

**Figure 5 pathogens-10-00142-f005:**
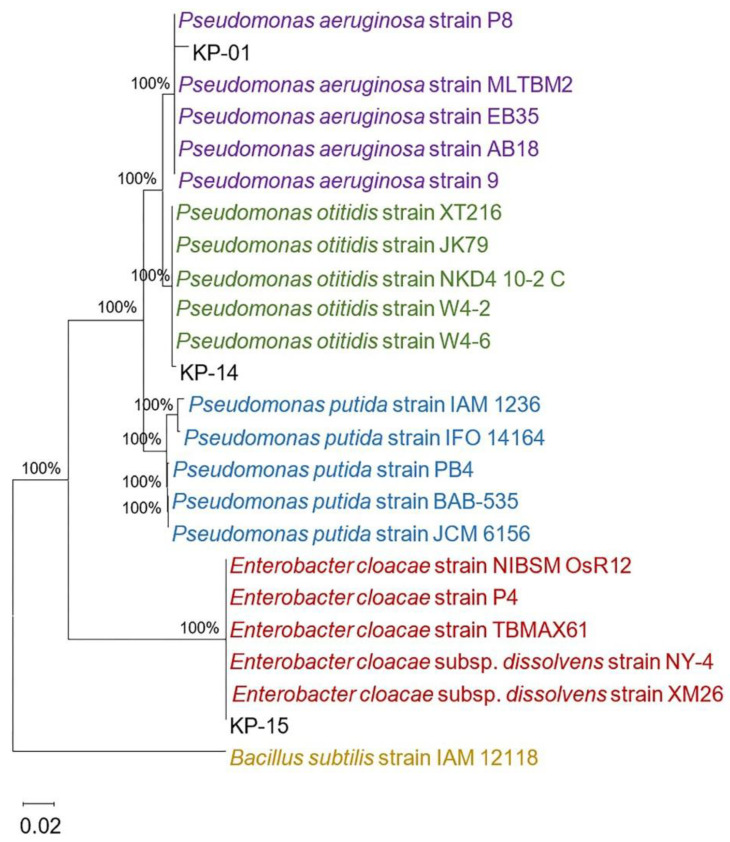
Phylogenetic tree showing the identity of partial 16S ribosomal RNA (rRNA) gene of three aquatic bacteria to other species from a database using MEGA version 10.0. The 16S rRNA gene sequences of three aquatic bacteria including KP-01 (MW498239), KP-14 (MW498240), and KP-15 (MW498241) (black alphabet) were phylogenetically analyzed in comparison to other *Pseudomonas* spp. (purple, green, and blue writing) and *Enterobacter cloacae* (red writing) using the neighbor-joining method of MEGA version 10.0. Bootstrap values for 1000 replicates are shown as a percentage at the nodes of the tree. *Bacillus subtilis* is used as an outgroup. The bar indicates 2% sequence divergence.

**Figure 6 pathogens-10-00142-f006:**
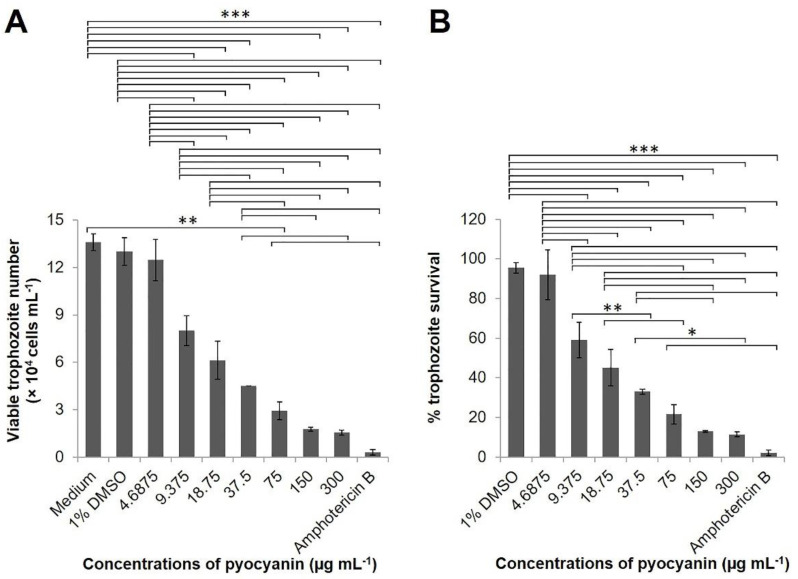
Anti-*Naegleria fowleri* effect of pyocyanin. Viable *N. fowleri* trophozoites at 24 h-post exposure with different concentrations of pyocyanin (4.6875, 9.375, 18.75, 37.5, 75, 150, and 300 µg mL^−1^), Nelson’s medium, Nelson’s medium containing 1% DMSO, and amphotericin B (10 µg mL^−1^). (**A**) Viable *N. fowleri* trophozoite number of 24 h post-treatment with pyocyanin as determined by trypan blue exclusion method. (**B**) Percentage survival of *N. fowleri* trophozoites at 24 h post-treatment with pyocyanin. Results are shown as mean ± standard deviation (SD) of the representative of three independent experiments. ∗, *p* < 0.05; ∗∗, *p* < 0.01, and ∗∗∗, *p* < 0.001.

**Figure 7 pathogens-10-00142-f007:**
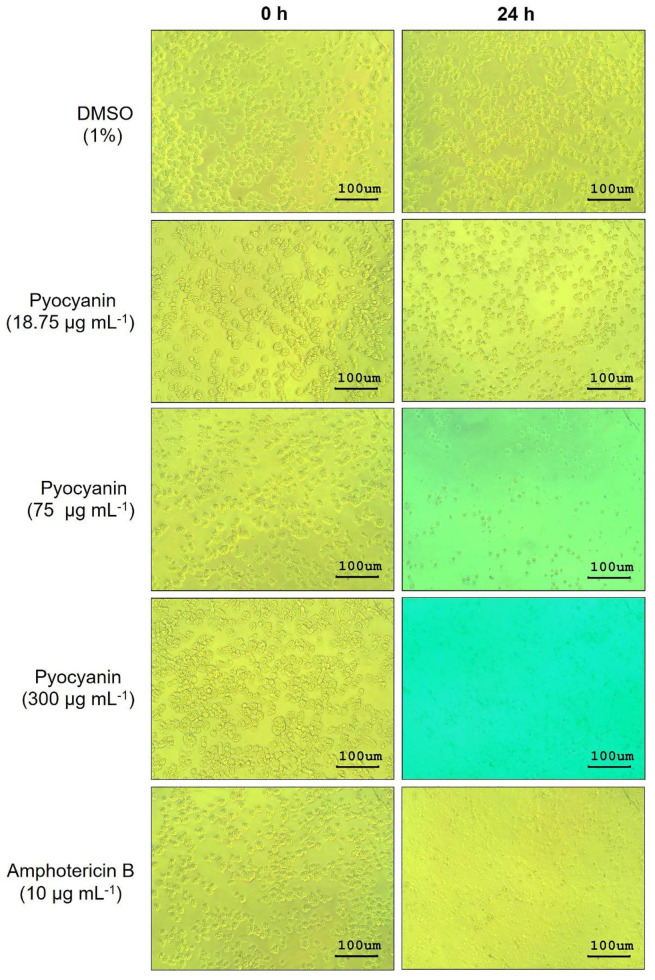
Microscopic morphology of pyocyanin-treated *N. fowleri* at different time intervals. *N. fowleri* (1 × 10^4^ cells) in 96-well plate were treated with different concentrations of pyocyanin (18.75, 75, and 300 µg mL^−1^). Medium containing 1% DMSO and amphotericin B (10 µg mL^−1^) were used as negative and positive controls, respectively. Morphological alterations and cell numbers were observed daily under 200× magnification.

**Table 1 pathogens-10-00142-t001:** Biochemical profile of the three unknown aquatic bacteria (KP-01, KP-14, and KP-15).

Biochemical Tests	KP-01	KP-14	KP-15
Glucose	−	−	+
Raffinose	−	−	+
Inositol	−	−	+
Urease	−	−	−
Lysine decarboxylase	−	−	−
Tryptophan deaminase activity (TDA)	−	−	−
Citrate	+	+	+
1,2,4,5-Tetrachlorobenzene (CL4)	−	−	+
Acetate	+	−	−
Sucrose	−	−	+
Rhamnose	−	−	−
Hydrogen sulfide (H_2_S)	−	−	−
Arginine dihydrolase	+	+	−
Esculin	−	−	+
Maltose	+	+	+
Nitrate	+	−	+
Sorbital	−	−	+
Arabinose	−	−	−
Indole	−	−	−
Ornithine decarboxylase	−	−	+
Voges–Proskauer	−	−	+
Ortho-nitrophenyl-β-D-galactopyranoside (ONPG)	−	−	+
Oxidase	+	+	−
Oxidative fermentation (OF)/glucose	+	+	+
Genus and Species Identification	*Pseudomonas aeruginosa*	*Pseudomonas putida*	*Enterobacter cloacae*

+, Positive; –, Negative.

## Data Availability

The data presented in this study are available on request from the corresponding author.

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
