# Peer review of "A Search for Anti-*Naegleria fowleri* Agents Based on Competitive Exclusion Behavior of Microorganisms in Natural Aquatic Environments"

_pathogens, 2021, doi:10.3390/pathogens10020142_

Round 1
Reviewer 1 Report
This study focuses on the search for an anti-N. fowleri agent produced by bacteria isolated from the natural aquatic habitat that might have potential application for the control of the pathogenic amoeba population in the environment.
From an experimental point of view it was well conducted. In my opinion, however, considering that Figures 3 and 6 are not very appreciated by the reader, it would be necessary to specify whether the treatment with CFSs and pyocinin also induces the passage of trophozoites into the cystic form.
Furthermore, in the results and discussion section, paragraph 2.3, it would be appropriate to also mention data regarding the toxicity of pyocyanin.
In fact, numerous studies have been undertaken assessing the cellular effects of pyocyanin exposure in various body systems, both in vitro and in vivo. A diverse range of toxic effects have been observed in animals and cells exposed to pyocyanin which have been shown to be predominantly mediated by free radical and pro-inflammatory cytokine production. Given the nature of pseudomonal infections, the effects of pyocyanin on the respiratory tract are the most widely studied phenomenon. Recent in vitro studies have, however, shown that pyocyanin has deleterious effects on cells in numerous other body systems, including the CNS, the urolological and the vascular system, with mechanisms other than oxidative stress implicated in a number of circumstances.
Author Response
Dear Reviewer 1,
Thank you for your generous comment on our manuscript. Please see the attachment.
Best Regards,
Pichet

Reviewer 2 Report
The manuscript "A search for anti-Naegleria fowleri agent based on competitive exclusion behavior of microorganisms in the natural aquatic environments" reports growth inhibition produce by cell-free culture supernatants. This work is of great interest in the field and in the control of this pathogenic amoeba in the environment.
- Manuscript elaboration. There are some grammar and punctuation mistakes that must be revised. Besides, some sentences may be rephrased to make them easier to understand. For example, line 18: “A sought for new effective…” This sentence should be rephrase, line 24 “The CFSs…”or line 35 “protozoon” ¿protozoa/protozoan?, among others throughout the text.
- Line 46 (references 7 and 8). Authors must include more recent citations to refer PAM cases. The ones included are from 1982 and 2008.
- Line 48. Same should be applied to reference 9. Try to include more recent citation.
- Please, include more information and references related to competitive exclusion in the Introduction section.
- Figure 1. How do you explain that no differences in trophozoites viability were observed comparing 1.20 and 2.40 mg/ml concentrations? Why this occurred?
- Line 95. Did you remove CFSs treatment after 48h incubation and incubate viable trophozoites in fresh medium? To test if remained viable trophozoites will be able to attach in these optimal conditions.
- Figure 6. Image quality must be improved. Please, make each image larger too, to see differences clearly.
- Line 228. Indicate DMSO concentration in this section (1%). It appears in line 181.
- Line 267. Why authors did not identify bacteria using PCR and sequencing?
Author Response
Dear Reviewer 2,
Thank you for your generous comment on our manuscript. Please see the attachment.
Best Regards,
Pichet Ruenchit

Reviewer 3 Report
Dear Authors,
I reviewed the manuscript Pathogens- 1066363. The authors of the submitted manuscript present some results regarding the identification of new anti-Naegleria fowleri agents. However, the article has several flaws in terms of text clarity and English spelling and should be fully reviewed in these respects.
I found a very limited number of papers dealing with the theme, and so I concede the novelty and contribution of this paper. There are, however, some authors who report that free-living amoebae (including N. fowleri), can be observed in association, namely with P. aeruginosa, and I would have liked to see some framework regarding these records (ex, https://pubmed.ncbi.nlm.nih.gov/10622622/).
It is necessary to clarify why point 2.3 appears rather abruptly (Effect of pyocyanin on N. fowleri). In other words, nowhere before is it explained why further tests are carried out with the compound pyocyanin and what is its relationship with the bacterial strains used. This compound was identified in some of the supernatants used in the assays?? If not, its use in additional tests lacks foundation and this should be fully clarified.
The document is globally clear but lacks the presentation of a solid foundation for the study's need and objectives. The bibliography review is limited and not very recent.
Revisions are needed, namely the following:
Line 48: Oceania
Line 83 and 84: The sentence “Proliferation of the amoebae was investigated using CellTiter 96® Non-83 Radioactive Cell Proliferation Assay.” should be moved to material and methods.
Lines 84: Remove “Panel”
Lines 85, 86, 87: Replace: “Panels B, C, and D 85 illustrate % survival of N. fowleri at 48 h post-treatment with different concentrations of the CFSs of 86 KP-01 (panel B), KP-14 (panel C), and KP-15 (panel D).” by “Survival rates (%) of N. fowleri at 48 h post-treatment with different concentrations of the CFSs of KP-01 (panel B), KP-14 (panel C), and KP-15 (panel D).”
Line 87: Rephrase “The data are…”
Author Response
Dear Reviewer 3,
Thank your for your generous comment on our manuscript. Please see the attachment.
Best Regards,
Pichet Ruenchit

Reviewer 4 Report
This manuscript describes the isolation of three bacterial strains that produce substances inhibitory to trophozoites of Naegleria fowleri. The paper is generally clear and well written but there are a few places where the English use is incorrect (see below). The description of the N. fowleri inhibition assay states that “Three independent experiments were done 251 and the results are shown as mean ± standard deviation (SD).” However, the error bars are all very small in every experiment conducted by this method given that they are derived from independent experiments? Were these data collected from triplicated wells in the same plate (in which case these are pseudo replicates) or do they actually arise from completely separate experiments? Can the authors confirm that the data are actually SD as stated and not SE?
The identity of the bacteria should be ascertained by 16S sequence analysis, although the identification of Pseudomonas aeruginosa and P. putida seems reasonable sure the identification of Enterobacter cloacae is not. PCR using 16S specific primers followed by sequencing of the products would remove uncertainty.
A major weakness of this paper is the use of the existing literature. In many cases the references cited are not at all suitable.
The first three references are historic and much more up to date reviews are available which would be of more use to readers
Reference 1 is a general (and old) review of pathogenic amoebae and is not suitable as used for Naegleria fowleri being an amphizoic amoebae that resides in water and soil.
Reference 3 is not a suitable reference for nostrils as a rout of infection, this paper is about histopathological changes caused by N. fowleri trophozoites in the grey matter of PAM patients, however it is suitable for the second citation.
Line 48. Neither reference 7 or 8 is correct as a source of the number of PAM cases reported world wide to date. Reference 7 gives a figure of 108 but this is paper dating from 1982. Reference 8 is more recent at 2008, and this paper does give a case number of 440 but cite Qvarnstorm et al, 2006 as a source for this number. The cited Qvarnstrom paper mentions 440 cases in the introduction but this figure is for cases from Acanthamoeba spp., Balamuthia mandrillaris and Naegleria fowleri. A better recent reference for the number of world-wide PAM cases is 431 from 2020 (Maciver et al, 2020 see below), or 381 (Gharpure et al, 2020 see below).
Line 154 Reference 19 is incorrect. This paper (Bottone et al, 2012) does not mention Pseudomonas aeruginosa.
Use of English.
Line 15 should be “humans” (plural) and again on lines 16 and 37. Also on line 16 “
choking water or inhaling dust” would be better as “inhaling water or dust”.
Line 18 The sentence beginning “A sought for new..” would be better as “
Seeking new effective and less toxic drugs for the environmental control of the amoeba population to reduce human exposure is logical for the management of N. fowleri infection”.
Line 24 change to “were seen to shrink and become rounded”
Gharpure, Radhika, et al. "Epidemiology and Clinical Characteristics of Primary Amebic Meningoencephalitis Caused by Naegleria fowleri: A Global Review." Clinical Infectious Diseases (2020).
Maciver, S. K., Piñero, JE & Lorenzo-Morales J. "Is Naegleria fowleri an Emerging Parasite?." Trends in parasitology 36.1 (2020): 19-28.
Author Response
Dear Reviewer 4,
Thank you for your generous comment on our manuscript. Please see the attachment.
Best Regards,
Pichet Ruenchit

Round 2
Reviewer 3 Report
Dear Pichet Ruenchit,
I revised the amended article and found that the changes made were in line with all the suggestions made, and overall they present a clearer and more robust article.
However, there are 2 points that need to be clarified:
1. What does "Results are shown as mean ± standard deviation (SD) of the representative of three independent experiments" mean? Apparently, you only present the results of 1 experiment, which you considered significant. Was this the procedure? If so, what is the criterion used to choose that same replica?
2. The additional data, now added, regarding the genetic analysis of bacterial species, in my opinion should be made clearer. Effectively the bacteria identified is Pseudomonas otitidis (99.71% identity) and not Pseudomonas putida. But there is some confusion, as the indication in table 1 (Biochemical profile of the three unknown aquatic bacteria), that it is Pseudomonas putida. In the present form, and as the results are shown, we have a clear contradiction between genetic and biochemical results. It is necessary to reformulate this part of the article, and eventualy repeat the biochemical analysis.
Author Response
Dear Reviewer 3,
Thank you so much for taking the time to assess our manuscript and kindly give a good suggestion. Please see the attachment. It included all points that we have addressed to the concerns you raised.
Best Regards,
Pichet Ruenchit

Reviewer 4 Report
The authors have addressed all concerns raised in my original report
Author Response
The Reviewer 4's comment is "The authors have addressed all concerns raised in my original report." So, we do nothing.